# The Effect of Physical Exercise on Fundamental Movement Skills and Physical Fitness among Preschool Children: Study Protocol for a Cluster-Randomized Controlled Trial

**DOI:** 10.3390/ijerph19106331

**Published:** 2022-05-23

**Authors:** Guangxu Wang, Yahua Zi, Bo Li, Shan Su, Lei Sun, Fei Wang, Chener Ren, Yang Liu

**Affiliations:** 1School of Physical Education and Sport Training, Shanghai University of Sport, Shanghai 200438, China; awangguangxu@163.com (G.W.); wangqiulibo@163.com (B.L.); 13653860771@163.com (S.S.); studentsunlei@163.com (L.S.); wangfei765265181@163.com (F.W.); chenerren@163.com (C.R.); 2College of Physical Education, Henan Normal University, Xinxiang 453007, China; 3School of Kinesiology, Shanghai University of Sport, Shanghai 200438, China; ziyahua@hotmail.com; 4Institute of Sports Science, Nantong University, Nantong 226019, China; 5Shanghai Research Center for Physical Fitness and Health of Children and Adolescents, Shanghai University of Sport, Shanghai 200438, China

**Keywords:** ball games, rhythm activities, basic movements, multiple activities, motor competence, health-related physical fitness, kindergarten, intervention

## Abstract

**Background**: Evidence shows that physical exercise promotes preschoolers’ fundamental movement skills (FMSs) and physical fitness (PF). However, studies that assess the effectiveness of different types of physical exercise interventions to improve FMSs and PF in preschool children remain scarce. To explore and compare the effectiveness of different physical exercise on FMSs and PF, interventions comprising ball games (BGs), rhythm activities (RAs), basic movements (BMs), and a combination of all related activities (multiple activities, MAs) will be conducted among preschoolers. **Methods**: A single-blind, five-arm, cluster-randomized trial will be conducted in kindergarten in Shanghai, China. In total, 300 healthy preschoolers, aged 4 to 5 years, will be randomized to four intervention groups (BG, RA, BM, or MA) and one control group (unorganized physical activities). Four intervention groups will receive three 30-min lessons weekly for 16 weeks. At the baseline, the end of the 16-week intervention, and the 6-month follow-up after the end of the intervention, the primary outcomes (FMSs and PF) and physical activity (PA), and sociodemographic and anthropometric data will be assessed. **Discussion**: This study will provide vital information regarding the effect of different physical exercise interventions on preschool children’s FMSs and PF, PA, and the potential interactions between these domains. The most effective intervention strategy can be generalized to kindergarten and other preschool educational institutions in practice to promote preschoolers’ development of FMSs and PF. **Conclusions**: This study protocol aims to provide a method to solve the problem of “how to arrange physical exercise and which kind of physical exercise program can promote FMS and PF better in preschool children”.

## 1. Background

Preschoolers are in a critical time of acquiring and consolidating fundamental movement skills (FMSs) [1] and developing physical fitness (PF) [2], which is essential to becoming physically active, less sedentary [3], and achieve health-related benefits both in the near future and in the long term [4,5,6].

Fundamental movement skills are defined as basic learned movement patterns that do not happen naturally and are the foundation of more advanced, complex movements required to participate in context-specific physical activities or sports [4,7]. FMSs include three domains: (1) locomotor skills, which involve locomotion of the body, e.g., running, jumping, etc.; (2) object control skills, which are mainly manipulative skills, e.g., throwing, catching a ball, etc.; and (3) stability skills, including static and dynamic balance [8]. Proficient FMSs may benefit children’s PF [9], weight status [6], social and cognitive capacity [10,11], and have other health-related benefits [12], and enable children to participate in physical exercises [5,13]. However, the evidence from previous study shows that 4.4% of young children (3–6-year-old) showed delayed development of FMSs and 8.8% were at risk of delay [14], and young children are struggling to achieve the recommended level of daily physical activity (PA) [15].

Physical fitness is a multifaceted construct involving physical and physiological components, including cardiorespiratory endurance, strength, speed, reaction, agility, balance, coordination and flexibility, etc. [16], and support individuals’ physical behavior, life, and work with vigor [9,17]. Research has proved that improvement of PF is important for the health of preschool children, particularly for obesity prevention [18]. Moreover, low levels of cardiorespiratory fitness and muscle strength in children are associated with low bone mass, metabolic risk factors, cardiovascular diseases, and premature death in older age [19,20,21]. PF and intellectual maturity are linked from an early age, and can even predict intellectual maturity in 3–6-year-old children [22]. Furthermore, small improvements in children’s fitness are accompanied by significant improvements in cardiovascular health [23,24]. Above all, improvements in PF are likely to be important for the prevention of cardiovascular events in early years [25].

Physical exercise is one of the critical ways to achieve, maintain, and improve FMSs and PF, and the association between PA and health outcomes is also mediated by FMSs and PF [26,27,28]. However, two recent systematic and meta-analysis reviews suggested that the results of a physical exercise intervention on PF among preschoolers should be interpreted with care because of the relatively small number of randomized controlled trials (RCTs) pooled in each variable and the quality of RCTs [25,29]. Another two reviews on interventions to promote FMSs in childcare and kindergarten also expressed concern about the interpretation of the effectiveness of interventions to improve FMSs due to the high heterogeneity of comprehensive evidence and the lack of long-term follow-up [30,31]. Furthermore, longitudinal studies that address the interaction between FMSs and PF among preschoolers are scarce. Many types of physical exercise were used to intervene in FMSs and PF in preschool children, such as physical games, rhythmic activities, sports, basic movement activities, aerobic training, and comprehensive activities [25,31]. Based on the current evidence, we conclude that well-designed, organized physical exercise programs can improve FMSs and PF better than free play or unorganized physical activities in preschool children [25,31]. A variety of programs have been provided for the promotion of FMSs and PF in preschool children, but the characteristics of the different forms of physical exercise mean that their physical effects are different [32,33,34]. The comparability between different physical exercise programs is low and inadequate; thus, it cannot be concluded which type of physical exercise results in better improvement of FMSs and PF in preschool children according to current evidence. This is not conducive for kindergartens to be able to choose appropriate physical exercise programs. One study found that physical exercise programs with various forms and functions have certain advantages over rhythmic gymnastics and football activities in improving FMSs [35]. Another study found that Sports, Play, and Active Recreation for Kids (SPARK) courses can improve FMSs better than gymnastics and conventional physical activities in preschool children aged 4–6 years [33]. Therefore, the development of physical exercise programs for preschool children should improve the target of interest. Moreover, the effects of different forms of physical exercise on the FMSs and PF of preschool children still need to be further studied.

In China, young children (3–6 years old) population is 57 million, 81.7% of whom were enrolled in kindergarten in 2018. Children spend approximately 40 h each week in kindergarten [36]. However, there is little empirical evidence showing the effect of physical exercise on preschoolers’ FMSs and PF in China. Above all, the present trial is designed to respond to the theoretical and evidence gap. Therefore, the aims of this trial are to (1) determine the effectiveness of each of these physical exercise interventions in improving preschool children’s FMSs and PF; and (2) establish the differences between the effectiveness of different physical exercise programs on preschool children’s FMSs and PF. The research hypothesis is that physical exercise interventions will improve children’s FMSs and PF more than unorganized activities, and that MA will show larger beneficial effects than BA, RA, or BM on children’s FMSs and PF.

## 2. Methods

### 2.1. Study Design

The study is a single-blind, cluster-RCT study, with the kindergarten class as the cluster for the intervention. This study is a part of the Physical Exercise on FMSs and PF of preschoolers (PEFP) project. The PEFP project is a series of studies, including a cross-sectional survey (covering six provincial and municipal regions in China) and the aforementioned RCT (implemented in Putuo District, Shanghai, China). Preschoolers recruited from 5 kindergartens will be cluster-randomly allocated to 5 groups (i.e., four intervention groups and one control group). The intervention groups are (1) the BG group, consisting of football, basketball games, etc.; (2) the RA group, including cheerleading, rhythm exercise, etc.; (3) the BM group, consisting of different basic movements, such as walking, running, jumping, climbing, throwing, catching, etc.; and (4) the MA group, combination of all kinds of physical exercise that have been used in other intervention studies. The control group (control) is an unorganized PA group (Figure 1). The content of the intervention comes from previous research combined with consultation and interviews with preschool educators [7,37].

### 2.2. Participants and Recruitment

The targeted population is preschool-enrolled children recruited through kindergartens. Eligible participants will be 4 to 5 years old and need to meet all the following criteria: (1) being healthy and (2) have the permission of their parent or guardian to participate in this study; written informed consent will be obtained. We will exclude individuals who have (1) severe cognitive impairment, (2) major medical or physical conditions affecting their participation in physical exercise, or (3) motor development delay. There will be no restrictions regarding the sex, ethnicity, or socioeconomic status of any of the participants.

### 2.3. Randomization and Masking

First, the information of the recruited kindergarten will be obtained, including the name of classes, and the number of children in each class. Then, an opaque envelope will be used for simple random assignment, and recruited kindergartens as a unit will be randomly assigned to one of five groups. The generation of the random assignment will be completed by the main person who is in charge of the intervention. As this is a behavioral and skill-specific intervention, eligible children in the same class will receive an identical strategy to avoid confounding influences, such as different exercise instruments or teachers. Furthermore, details of the intervention group allocation will not be available to participants and their teachers. Data analysts will not include blind processing, which will be performed by two separate individuals for statistical analysis of the research results. Participants and teachers will be unblinded when the research is finished by providing a simple report and brochure to the kindergarten.

## 3. Interventions and Procedures

### 3.1. Intervention Groups

#### 3.1.1. Development of Interventions

The intervention procedures have been developed by (1) applying the framework of exercise programs proposed by the American College of Sports Medicine (ACSM) [38]; (2) constructing domain-specific content for physical exercise interventions via a review of the existing literature for young children [7,26,27,31,37]; (3) obtaining open-ended responses concerning how preschool children exercise to improve FMSs and PF from preschool education experts and teachers; (4) constructing the actual intervention procedures; and (5) examining the validity of factor structures underlying the intervention procedures by consulting other preschool education experts and teachers and conducting pilot studies.

A review of previous studies shows that physical games (aerobic games, ball games, etc.), rhythmic activities, sports (basketball, football, swimming, etc.), fundamental movements activity (gross motor movements, functional movements, etc.), aerobic training, and comprehensive physical exercise were used in FMSs and PF intervention programs for children aged 4–5 years, and proved to be superior to free play or unorganized physical activity [25,29,31]. The results of interviews with preschool education experts and teachers, and the restrictions of the Ministry of Education of the Chinese government on the implementation of standardized “curriculum” and competitive sports in kindergartens were considered [39]. Ball games, rhythmic activities, basic movements, and multiple activities are the physical exercise programs of this study.

#### 3.1.2. Contents of Interventions

Each of the four interventions is based on structured lessons, which will involve a 30-min lesson three times per week for 16 weeks. In all four groups, each lesson will consist of a 5-min warm-up, 20 min of core exercises, and a 5-min cool-down activity. The warm-up will consist of light-intensity movements (e.g., wrist rotation and leg swing), moderate-intensity activities (e.g., arm rotation and knee-up walk), and higher-intensity activities (e.g., arm sprint and on-site running); cool-down will start with moderate-intensity activities and end with light-intensity movements. The main part of the lesson will include 5 min of moderate-to-vigorous intensity activity according to the respective intervention procedures, with at least 2 min of vigorous-intensity activity, every 10 min [40,41]. The four interventions will be carried out in the form of games to increase children’s interest, and will only differ in the main part (Table 1). The intervention will be carried out within the PA plan of kindergartens, to avoid extra PA for the preschool children in the intervention groups.

### 3.2. Control Group

To avoid the impact of activity experience on the FMSs and PF of preschool children in the control group, preschool children in the control group will participate in “unorganized PA”, while the intervention groups will participate in the designed lessons. The unorganized PA in this protocol is defined as: (1) children perform PA without the instructions of teachers, who are only responsible for the safety of children; (2) the types and intensity of activities are not affected by the teachers; (3) competitions that are spontaneously organized by young children, such as racket, dribbling, and balance beam competitions, are also unorganized PA. The PA content of the control group will be reported by their teacher once a week to ensure that the PA content of preschool children in the control group meets the requirements of the intervention program.

#### 3.2.1. Delivery and Fidelity Check of Interventions

Before the beginning of the formal intervention study, a workshop for PEFP members and a pilot study will be carried out to examine (1) the contents of the training program for interventionists and (2) the feasibility of intervention programs.

The formal physical exercise intervention will be implemented by kindergarten teachers participating in this study, and teachers will receive a two-hour training before the intervention at the kindergarten or university location. The training will be carried out with a prepared intervention portfolio, presentations, and visual materials. Following the training, a guided interview will be conducted by a second researcher to obtain information regarding (a) teaching and mediation competence using methods and materials, with consideration of the preschoolers’ characteristics; (b) whether the interventionist will implement the interventions as planned; and (c) any comments on improvements in the training and general remarks.

In the pilot study, each interventionist will carry out the physical exercise intervention lessons according to the randomized group and provide feedback on the feasibility of the lessons according to different criteria (time, content, goals, etc.) using a checklist provided by the researchers. Each intervention group will be assigned an additional researcher to supervise the quality of the intervention. In addition, the researcher will also play a protective role and record any special situations (accidental injuries) that occur during the intervention.

In addition, the PEFP members will also keep in close touch with preschool teachers via instant networking software. So, throughout the execution of the study, we will share information and solve potential problems in a timely fashion. First, files and videos of the intervention procedures will be distributed to class instructors or teachers ahead of time to remind them of the contents that are going to be implemented. Second, the teamwork group will help solve the problems encountered online.

#### 3.2.2. Conditions for Termination of Intervention

The injury risk associated with the intervention content in this study is low, and few injury events have been reported in similar studies [36]. During implementation, unless injured child(ren) are affecting the conduct of the intervention or other irresistible factors, the intervention will continue as planned. The intervention and data analysis will be conducted according to an intention-to-treat analysis. The data of all the enrolled participants will be analyzed; the missing data will be processed using the mean interpolation method. If injury to a child occurs during the course, the kindergarten, parents, and research leader will jointly negotiate the treatment plan. The FMSs and PF of all subjects will be assessed before and after the intervention, and at the end of the 6-month follow-up. The results of the follow-up will be used for auxiliary analysis. Consent to participate in this program will be obtained from the recruited children’s parents, and the children can quit at any time. If the intervention is terminated due to irresistible external factors (such as suspension of classes due to COVID-19), the intervention duration will be extended or another time will be chosen to restart the intervention according to whether the intervention is terminated for more than 2 weeks. All decisions will be made by the project leader.

#### 3.2.3. Measurement Slots

Participants will be assessed three times: at baseline, the end of the intervention, and the end of the 6-month follow-up. At each time point, we will assess participants’ FMSs, PF, PA, and descriptive data (such as age, sex, weight, height) within one week. The executive plan of FMSs and PF is outlined in Figure 2.

### 3.3. Measurements

Primary and secondary outcome measures, and descriptive measures will be assessed in the three time slots. To reduce the loss of participants, researchers will inform teachers about the details of the measurements before the measurements start. Teachers will communicate with parents to avoid unnecessary loss of preschool children.

#### 3.3.1. Primary Outcome Measures

The primary outcomes in this study are FMSs and PF.

##### FMSs

Children’s FMSs will be assessed by a battery of tests from the Test of Gross Motor Development (TGMD) [42] and the Movement Assessment Battery for Children (MABC) [43]. The domain-specific content of FMS assessment was constructed in three phases. Phase 1 focused on the existing mature instruments for measuring FMSs. Three sub-domains represent FMSs competence: (a) locomotor skills, (b) object control skills, and (c) stability skills [8]. Within these subdomains, seven factors were postulated, comprising locomotor skills (i.e., 10-m shuttle run and hopping), object control skills (i.e., bouncing a ball, jamming coins, and kicking), and stability skills (balance beam and one-leg stand). Phase 2 involved a review of the existing FMS literature to identify each of the three subtest constructs. Phase 3 involved the use of a panel of experts to evaluate the FMSs measurements and consisted of a series of rounds before a final consensus is reached by the panel. After two rounds of the three-phase process, the three general domains representing FMSs were identified. The final pool of FMS assessment items includes measurements of running and hopping for locomotor skills; bouncing a ball, jamming coins, and kicking for object control skills; and the balance beam (for dynamic balance) and one-leg stand (for static balance) for stability skills (Table 2).

##### PF

We will use PF test kits (The PREFIT Battery) [2,44], which has been widely recognized and applied [2,45,46]. The PREFIT battery consists of measurements of the balance beam walk for assessing balance ability; handgrip and standing long jump for upper and lower limb strength, respectively; sit-and-reach for flexibility; 10-m shuttle run for agility; hopping for coordination; and 20-m shuttle-run for cardiorespiratory fitness (Table 3).

#### 3.3.2. Secondary Outcome Measures

The secondary outcome measure in this study will be PA.

PA will be assessed objectively using GT3X+ accelerometers (Pensacola, Fla., USA) and subjectively using the Preschool-age Children’s Physical Activity Questionnaire (Pre-PAQ) [47,48].

For the accelerometry, preschool children participating in this study will be instructed to wear the accelerometers over their waist consecutively for seven days, and will stop wearing them when they go to bed and during bathing and water activities. Teachers, parents, and preschool children will be instructed to take care of the devices and log the time when the devices are removed. Before being distributed to the children, the accelerometers will be set to record activity counts in 3 s epochs [49].

For the subjective self-reporting, we will use Pre-PAQ, which is a 37-item 3-day parent-reported questionnaire measuring the regular PA and sedentary behaviors of the parent and the preschool-aged child in the home environment. The Pre-PAQ has been shown to have acceptable validity and reliability in a sample of parents of preschool-aged children [50]. PA intensity and sedentary activity minutes will be analyzed separately according to the weekday or weekend.

### 3.4. Descriptive Measures

The descriptive measures in this study will be (1) sociodemographic characteristics; and (2) anthropometric data, including height, weight, and body mass index (BMI).

#### 3.4.1. Sociodemographic Characteristics

Parents will be instructed to complete a questionnaire about their child’s information including the date of birth and sex, household factors, family’s socioeconomic characteristics (family income, parental and primary guardian education, and parental employment status), child’s health status (reporting medical conditions and medications, if any), and child’s sleep time.

#### 3.4.2. Anthropometric Data

Standing height will be measured barefoot using a stable stadiometer (GMCS-SGZG3, Jian-Min, Beijing) to the nearest 0.001 m. Bodyweight with light clothes will be measured using a portable scale (GMCS-YERCS3, Jian-Min, Beijing) to the nearest 0.1 kg (kg). BMI was calculated by height and body weight:$$\mathrm{BMI}\;\left(\mathrm{kg}/m^{2}\right)=\frac{\mathrm{Weight}\;\left(\mathrm{kg}\right)}{{\left(\mathrm{Height}\right)}^{2}\;\left(m^{2}\right)}$$

## 4. Statistical Analysis

### 4.1. Sample Size

The primary outcome variables in this study are FMSs and PF. Before recruitment, a priori power analysis was performed in G*Power 3.1 to calculate the sample size necessary to detect meaningful changes from the baseline to the post-intervention time points in the total raw scores of seven FMSs (i.e., 10-m shuttle-run, hopping, bouncing a ball, jamming coins, kicking, balance beam, and one-leg stand) and seven PF (i.e., balance beam, handgrip, standing long jump, sit-and-reach, 10-m shuttle running, 20-m shuttle running, handgrip strength, and hopping). Using an alpha of 0.05 and a power of 80%, the calculation of the power to detect a pre-post change in these measures was based on the effect sizes reported in a recent systematic review. The expected effect size of the intervention on FMSs and PF was at 1.42 and 0.36 (standardized mean difference, SMD), respectively [51]. With reference to these results, a prudent estimation of the effect size (SMD = 0.36) was used for the calculation. The required sample was calculated to be 70. To account for the clustering effect, this number was then multiplied by a design effect (DE) of 1 + (*m* − 1) × ICC, where *m* is the average cluster size and ICC was estimated at 0.05. With an estimated average class size of 30, at least 150 participants will have to be recruited. Based on previous studies, we presumed a drop-out rate of 15%, implying a minimal sample size of 150/0.85 = 176 children in total. This study takes the class as the cluster of random sampling; the class size in recruited kindergartens is 30–35. Therefore, considering the drop-out rate and special samples (preschool children with developmental retardation or hyperactivity disorder), each experimental group was divided into two classes, implying about 60 children in each group. The total size is 300.

### 4.2. Data Collection

The test will be conducted by members of the PEFP research group, who have no competing interests, and details of the intervention group allocation will not be available to the testers. Before the test, two training sessions will be conducted to ensure the quality of the test, which will be conducted by professionals from the test equipment company. The workshop will (1) cover the development characteristics and special needs of young children; and (2) deliver the basic concepts, testing skills, and emergency plans for young children.

### 4.3. Data Management

All research data entry will be carried out by two people, followed by an interrater consistency check. Consent forms will be stored separately from participant data, and a unique identifier code will be assigned to each participant. Data will be deleted immediately from voice recorders after the transcription, with pseudonyms used in all reports in place of participants’ names. Data will only be accessible to primary researchers and statisticians after application to the research group. Data will be stored for a maximum of 5 years before being securely destroyed.

All phases of this study will be monitored by a data safety officer, who has worked for monitoring data collection and data analysis. The data safety officer is responsible for the data management and distribution of applied data, and will not obtain the details of the research. Management of the trial data will be in accordance with the data management plan, which was developed and approved by the project group. Moreover, the data will be published in two ways: writing academic papers and uploading to the experiment registration website.

### 4.4. Data Analysis Plan

Baseline characteristics and unadjusted study outcome measures will be summarized by intervention groups using descriptive statistics and used to assess between-group equivalence at baseline. Prespecified baseline covariates in both primary and secondary outcome analyses will include age, sex, FMSs capacity, PF level, and PA level.

Baseline demographic descriptors and primary and secondary outcome measures will be compared across groups by using analysis of variance (a non-parametric test will be used when the data does not conform with a normal distribution) for continuous variables and the χ^2^ test for categorical variables. The planned descriptive data on measured-point FMSs and PF will be tabulated across the intervention groups and control group. In our primary analysis of the FMSs and PF outcome, we will use mixed model analyses to estimate the differences with their corresponding 95% CIs, comparing the BA, RA, BM, and MA groups with the control group. Time, physical exercise programs (intervention or control groups), and their interaction will be fixed factors. The random effects will be the kindergarten and individual to take into account the clustered data structure.

Multilevel linear regression models will be constructed to assess the effectiveness of the physical exercises on FMSs and PF performance. The baseline data of the outcome variables, assessment points, intervention groups, and any imbalanced covariance will be added to the model along with the kindergartens and PA as random effects. The interaction between physical exercise and FMSs and PF will be tested.

Secondary analyses will investigate the effect of the intervention on other domains using different types of regression analyses, depending on the outcomes. Logistic regressive analyses will be constructed to assess the interference factors. The models will be used to assess the effect of forms of physical exercise, and to explore associations between secondary outcome measures, and will be adjusted for various confounding factors (e.g., sex, PA).

## 5. Ethics and Dissemination

An informed consent form will be provided to all participants. Children’s informed consent form will be filled in by their primary guardian, and the preschool administrators will be instructed to sign the corresponding consent. All informed consent forms will be written and recovered and kept by the data manager. Children’s willingness to participate in the study will be determined by communication between the teachers and preschool children. The teacher will introduce the content of the study and the work that the children will need to do, and ask the preschool children if they would like to participate. The intervention will only start after the consent of the preschool children is obtained. Permission for the research protocol was obtained from the ethics review committee of Shanghai University of Sport before participants recruitment.

## 6. Discussion

A panel of researchers has recommended that policies be implemented to increase PA and improve FMSs proficiency and PF levels in preschoolers through school-based initiatives in China. However, the scarcity of physical education lessons specific for preschoolers limits their opportunities to acquire FMSs capacity and meet PF standards. The interventions in this study have several significant points. They use existing school sports facilities (e.g., football, basketball) or even no sport devices (in rhythm activities) to arouse children’s interests to learn and practice motor skills and improve their PF level, without overburdening parents and teachers. The interventions do not require any changes to the curriculum. Overall, the PEFP program represents an attractive strategy for improving FMSs and PF levels among preschool children, providing motor skill-related activities through structured in-school time.

Although there is existing evidence regarding the effectiveness of programs to improve FMSs and PF in healthy preschoolers, this needs to be interpreted prudently as it is based on low-quality evidence, the intervention strategies are combined with other factors (e.g., nutrition, environment), and immediate post-intervention effects without long-term follow-up were assessed [7,36,37,38]. Moreover, in China, except for a policy-driven study on early childhood PF promotion [4], there is a lack of empirical studies concentrating on improving FMSs and PF using school-based physical exercise program interventions. It is, therefore, imperative that the effectiveness of school-based interventions is established to promote FMSs and PF in Chinese children during the vital window in time for FMS acquisition and PF development. In addition, the cohort identified for this school-based physical exercise program intervention will provide possibilities for future research on the influence of early childhood factors on future habitual PA, motor proficiency, and longer-term health outcomes.

## 7. Challenges and Limitations

The current protocol has some challenges. First, the participants in this study are 4-–5-year-old preschool children who still largely lack self-control and attentional focus. It will, therefore, be difficult to engage them continuously in the designed physical exercise program. To meet this challenge, the research group will staff each intervention group with two researchers to ensure preschool children’s safety and the intervention quality. Second, the intervention will last for 16 weeks, which will place a great human resource pressure on the PEFP group. We plan to stagger the intervention time of two close kindergartens, and the intervention will be carried out between 8:30 and 11:30 in the morning. In this way, the researchers assigned by the PEFP group can assist more intervention groups within the time allowed.

There are some limitations to this protocol. First, young children are in a rapid period of growth and development; thus, it will be difficult to identify whether natural growth or the exercises intervention contributes to the improvement in children’s FMSs and PF. Like other studies, this protocol will set up a control group in the same school with each intervention group to reduce the confounding impact of natural growth as much as possible. Second, the intervention in this study will last 16 weeks, which seems insufficient to improve FMSs and PF. However, the duration of FMSs or PF interventions in previous studies varied from eight weeks to one academic year or longer [32,41,52,53].

## 8. Conclusions

This study protocol aims to provide a method to solve the problem of “how to arrange physical exercise and which kind of physical exercise program can promote FMS and PF better in preschool children”.

## Figures and Tables

**Figure 1 ijerph-19-06331-f001:**
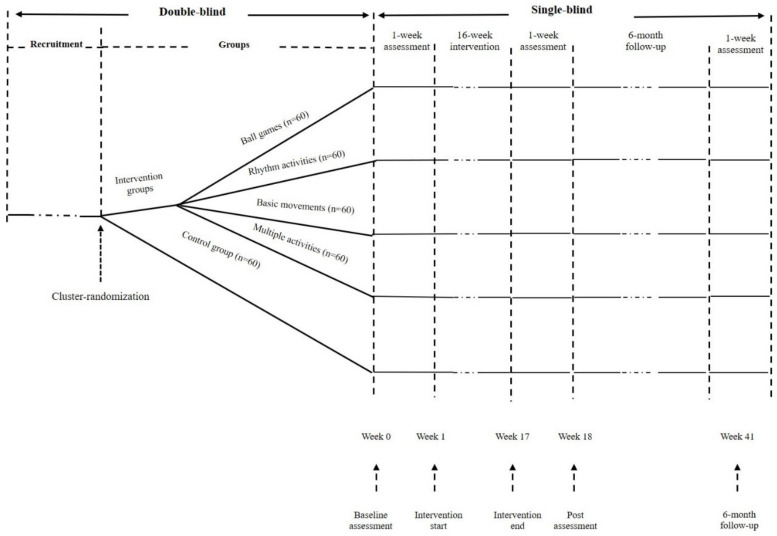
Design of the PEFP study.

**Figure 2 ijerph-19-06331-f002:**
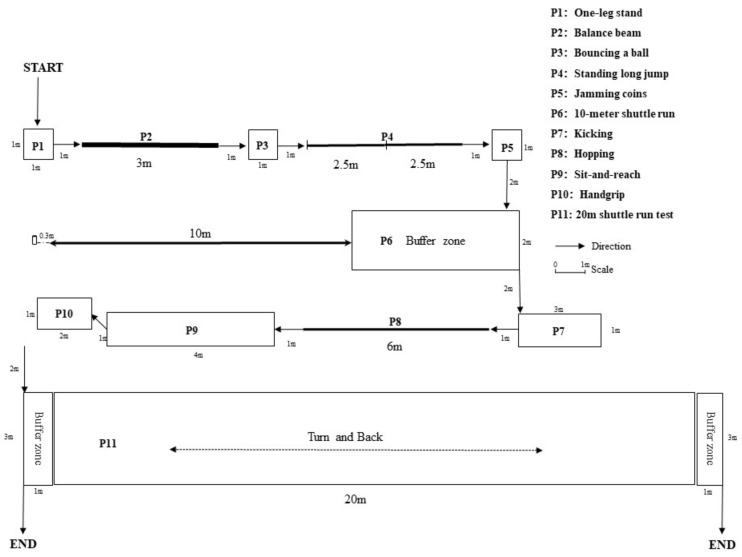
Flow of the assessments of measures in the trial.

**Table 1 ijerph-19-06331-t001:** Description of the contents, specific focus, and time allocation of the four interventions.

Interventions	Specific Focus	Themes	Contents	Lessons Needed
Basic movements	Aims to help preschool children learn and master FMSs, and improve muscular strength, coordination, and sensitivity.	Stable skiers (aiming at stability skills)	Standing and walking	10
		Happy little animals (locomotor skills)	Climbing and running	10
			Jumping	10
		Little ball players (object control skills)	Throwing/catching a ball	6
			Slapping ball	6
			Rolling, kicking, and dribbling	6
Ball games	Enable children to learn and master the basic ball skills, and at the same time promote children’s fitness level.	Rubber ball games	Running/jumping with ball	1
			Rolling pass, throwing, and catching	6
			Slapping and pitching	5
		Basketball games	Slapping basketball	2
			Dribbling	3
			Passing and catching	4
			Throwing, catching, and shooting	4
		Football games	Kicking, stepping over	7
			Dribbling and small-sided games	5
		Tennis games	Tapping	2
			Throwing and catching ball	6
			Hitting fixed ball	3
Rhythmic activities	Develop rhythmic gymnastics, aiming to improve physical control, aerobic fitness, and shaping grace posture.	Animal imitation exercise	Learning basic posture	4
			“I’m a little pacesetter”	12
		Baby warm-up exercise	Basic and jump steps, and body wave	8
		Pizza	“Little swan”	12
		Cheering on	Flower ball cheerleading	12
Multiple activities	Integrates theme activities into physical exercise to cultivate children’s interest and habits regarding regular exercise	Traditional Chinese games	Throw handkerchief games	4
			Beanbags, jumping rope, bench and coin games, etc.	8
		Animal imitation	Hawks Catch chicken, etc.	12
		Role playing and cooperative practice	Pushing cart game, etc.	12
		Military games	Brave little soldiers, etc.	12

**Table 2 ijerph-19-06331-t002:** The Fundamental Movement Skills Tests.

FMS Proficiency	Test	Materials and Site Layouts	Directions	Performance Records	Cautions
Locomotor skills	10-m shuttle run	Materials: flexible rulers, marking tap, poles, and stopwatch Site layouts: two parallel lines 10 m (m) apart will be marked on the floor using marking tap	Child runs concurrently with an audio signal, then runs back once reaching the turning line	The result will be recorded in seconds	Stay clear of the starting and turning line
Hopping	Hopping	Materials: tape measure, marking taps, stopwatch, and beanbag 10 cm long, 5 cm wide, and 5 cm high Site layouts: 10 parallel lines 0.5 m apart, on which a piece of beanbag will be placed	Child stands behind the starting line and hops with close feet continuously as soon as he/she hears the audio signal, and until he/she hops over the last beanbag	The result will be recorded in seconds	Repeat the test if the child hops over the beanbag with one leg, steps onto the beanbag, or kicks the beanbag from where it is two times
Object control skills	Bouncing a ball	Materials: soft balls and stopwatch. Site layouts: 1 m × 1 m square	Child dribbles the ball with one hand 8 times (as ball bouncing higher over his/her knees, it is counted as one)	The result will be recorded in seconds	Push balls with fingertips, not a slap
Object control skills	Jamming coins	Materials: 8 coins, a box with slot on the upper side and stopwatch	Child stuffs 8 coins into the box through the slot with the dominant hand	The result will be recorded in seconds	
Object control skills	Kicking	Materials: Sand-filled solid balls and sign barrels Site layout: 3 m, one sign barrel every 1.5 m	Child kicks the solid ball from one side to the other side with the arch of their foot, bypassing the 1.5-m sign barrel, and the route is in the shape of “C”	The result will be recorded in seconds	Repeat the test if child kicking the ball not with arch of foot
Stability skills	Dynamic balance (Balance beam)	Materials: balance beam (3 m long, 10 cm wide, and 30 cm high) with two platforms (20 cm long, 20 cm wide and 30 cm high) attached to both ends, and a stopwatch	Child walks consecutively forward on the balance beam, until one of the feet touch the platform on the other side	The result will be recorded in seconds	1. Repeat the test if the child falls off the beam 2. Testers should protect the child from harm during the test
Stability skills	Static balance (One-leg stand)	Material: stopwatch	Child stands on one foot with the supporting leg on the floor and the free leg flexed at the knees, maintaining the balance posture until the required posture cannot be maintained	The result will be recorded in seconds	

**Table 3 ijerph-19-06331-t003:** Physical Fitness Tests.

PF Proficiency	Test	Materials and Site Layouts	Directions	Performance Criteria	Cautions
Balance	Balance beam	As aforementioned in Table 2.			
Upper Limbs Strength	Handgrip	Materials: grip meter special for young children (TKK5825 TKK, in TAKEI, Japan).	Child squeezes gradually and continuously for at least 2 s or 3 s with one hand.	The result will be recorded in kilograms.	Testers adjust the grip meter for the optimal span for the child before the test.
Lower limbs strength	Standing Long Jump	Materials: tape measure, marking taps. Site Layout: a take-off line marked on the surface with taps.	Child will be instructed to stand with feet at shoulder width, and toes just behind the take-off line; then, child bends knees while swinging both arms, and pushes off vigorously and jumps as far as possible.	The reported distance is measured from the take-off line to the point where the back of the heel nearest to the take-off line lands on the ground. The result will be recorded in meters.	
Flexibility	Sit-and-reach	Material: sit-and-reach device.	Child will be instructed to take off their shoes, sit on the floor with legs fully extended, and press his/her feet against the base of the device. With arms fully extended and hands overlapped with palms down, child reaches as far forward as possible and pauses for 2 s.	The result will be recorded in centimeters.	Child will be required to stretch his/her lower back, hamstrings, and calves before the test.
Agility	10-m shuttle-run	As aforementioned in Table 2.			
Coordination	Hopping	As aforementioned in Table 2.			
Cardiorespiratory fitness	20-m shuttle-run	Materials: flexible rulers, marking tap, poles, and stopwatch. Site layout: two parallel lines 20 m apart will be marked on the floor using marking tap.	Child will be instructed to run progressively to exhaustion from one line to another 20 m apart, changing direction at the pace set by an audio signal that increases progressively.	The test ends when the child fails to reach the end lines concurrent with the audio signal on two consecutive occasions or when the child stops because of exhaustion.	Testers will be continually encouraging participants so that they give their best performance. Child performs a test no longer than 8–10 min.

## Data Availability

Data sharing is not applicable to this article as no datasets were generated or analyzed during the current study.

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
