# Peer review of "The Effect of Physical Exercise on Fundamental Movement Skills and Physical Fitness among Preschool Children: Study Protocol for a Cluster-Randomized Controlled Trial"

_ijerph, 2022, doi:10.3390/ijerph19106331_

Round 1
Reviewer 1 Report
The paper presents study protocol for the research on the effect of physical exercise on fundamental movement skills and physical fitness among preschool children. The research was well designed. The size of the groups of children was well defined. Preschoolers recruited from 5 kindergartens will be cluster-randomly allocated to 5 groups (i.e., four intervention groups and one control group). The aim is to look at the effectiveness of different types of activities. Well described is control group and the protocols are well defined. The fundamental movement skills tests and physical fitness tests are well described. Challenges and limitations are presented, which is very important in this type of research. English language is correct.
Similar studies were conducted, but the research was not well defined and this paper is filling the gap. I think that this study protocol can be published in presented form.
Author Response
Response: Thank you very much for your recognition. There are many studies on the impact of sports activities on FMS and PF of preschool children. However, many physical exercise programs in these researches cannot be popularized in kindergartens. For example, competitive sports are not allowed to be used as the teaching content and difficult to become the regular curriculum of kindergartens in many countries and regions like in China. Many physical exercise programs in current studies are difficult for kindergarten teachers to implement. There are few studies take teachers as implementers to design physical exercise plan. As a result, how to arrange physical exercise and which kind of physical exercise can promote FMS and PF better in preschool children, and can be implemented by kindergarten teachers without too much difficulties are still unclear. Therefore, we designed this study protocol, hope can fill these research gaps to a certain extent after implementation.

Reviewer 2 Report
The study proposes a study to examine the effects of different types of physical exercise on fundamental movement skills and physical fitness among preschool children. Overall, I think this study is important, the manuscript is well-written and the protocol is reasonable, and I believe that the results will be useful to develop physical exercise effects on fundamental movement skills and physical fitness. Below are more specific comments that need to be considered.
The study selects 5 types of intervention, namely ball games, rhythm activities, basic movements, multiple activities (16 physical activities), and control (unorganized physical activities) interventions. However, it is unclear why these 5 interventions are selected. One of the main goals of the study is “the characteristics of different forms of physical exercise determine that their effects on the physical are different. The effects of different forms of physical exercise on FMS and PF of preschool children still need to be further studied. (line 72-74)”. It is thus important, first, to more clearly review the current literature to understand the intervention effects on different types of exercise to guide the selection of exercise type for the current study. Second, some hypotheses could be formed based on the literature that why some types of intervention are particularly appropriate (or effective) for preschool children (4-5 years old). For example multiple activities intervention could be particularly effective due to regularly changing activities, so it is more fun for kids (not do the same activity every time etc). Or rhythm activities are better because an increase in cardio responses is moderate, compared to ball games etc. Or ball games are better because of the involvement of the competition component. There are many possibilities and a hypothesis can be derived to be tested in the current study. The authors can elaborate more on different types of exercise intervention effects and come up with their own hypotheses.
As mentioned by the authors, some major limitations of previous studies are smaller sample sizes and lack of long-term follow-up. The current study proposes 60 subjects in each intervention group with one follow-up after 6 months of the intervention. I am not sure whether this sample size and follow-up plan are strong enough to solve the questions raised by the literature. Although I absolutely agree that this will already be a useful and important study, the sample sizes and follow-up plan, from my perspective, are not adequate to solve the aforementioned issues.
5 kindergartens will be randomly allocated to 5 intervention groups. So, the study will compare the effects of 5 different kindergartens. I am not sure whether this could be problematic, as kids in different kindergartens could be very different, the quality of PA teachers in different kindergartens could also be different, potentially resulting in some differences in the tested results (FMS and PF), or even in the baseline results. It is more idea to compare in the same kindergarten, if possible, or to include more kindergartens (not only 5 kindergartens).
Minor notes
…and largely determines PA level… (line 49), it is weird that physical fitness determines physical activity? It is better to say that physical fitness is influenced by physical activity.
the effectiveness of interventions to improve FMS due to the quality of evidence…(line 66-67), this can be more specific and clear, so the readers know what do you mean “the quality of evidence”, e.g., what evidence showing poor quality of the results or…
(2) establish the comparative effectiveness of these physical…(line 82) seems to indicate the same thing as (1) determine the effectiveness of each of these physical exercise interventions...
Author Response
- The study selects 5 types of intervention, namely ball games, rhythm activities, basic movements, multiple activities (16 physical activities), and control (unorganized physical activities) interventions. However, it is unclear why these 5 interventions are selected. One of the main goals of the study is “the characteristics of different forms of physical exercise determine that their effects on the physical are different. The effects of different forms of physical exercise on FMS and PF of preschool children still need to be further studied. (line 72-74)”. It is thus important, first, to more clearly review the current literature to understand the intervention effects on different types of exercise to guide the selection of exercise type for the current study. Second, some hypotheses could be formed based on the literature that why some types of intervention are particularly appropriate (or effective) for preschool children (4-5 years old). For example multiple activities intervention could be particularly effective due to regularly changing activities, so it is more fun for kids (not do the same activity every time etc). Or rhythm activities are better because an increase in cardio responses is moderate, compared to ball games etc. Or ball games are better because of the involvement of the competition component. There are many possibilities and a hypothesis can be derived to be tested in the current study. The authors can elaborate more on different types of exercise intervention effects and come up with their own hypotheses.
Response: Thanks for your suggestion, and we also accept your suggestions very sincerely. We have added reviews of the current literature and the intervention effects on different types of exercise, “Based on the current evidence, we conclude that the well designed, organized physical exercise programs can improve the FMS and PF better than free play or unorganized physical activities in preschool children”, please check line 71-74. And “The comparability between different physical exercise programs is low, and inadequate to conclude which type of physical exercise can improve the FMS and PF better in pre-school children according to current evidence, that's not conducive for the kindergarten to choose appropriate physical exercise programs. One study found that physical exercise program with various forms and functions have certain advantages over rhythmic gymnastics and football activities in improving FMS. Another study found that Sports, Play, and Active Recreation for Kids, SPARK courses can improve the FMS better compared with gymnastics and conventional physical activities in preschool children aged 4-6 years. Therefore, the development of physical exercise programs for preschool children should improve the interest.”, please check line 76-85. Along with part "Development of interventions" line 154-163, can help to explain why these 5 interventions are selected and how the current hypotheses come up.
- As mentioned by the authors, some major limitations of previous studies are smaller sample sizes and lack of long-term follow-up. The current study proposes 60 subjects in each intervention group with one follow-up after 6 months of the intervention. I am not sure whether this sample size and follow-up plan are strong enough to solve the questions raised by the literature. Although I absolutely agree that this will already be a useful and important study, the sample sizes and follow-up plan, from my perspective, are not adequate to solve the aforementioned issues.
Response: Thank you very much for your concern. I must admit that our research design is not perfect. As we summarized in the manuscript, there are many limitations in the current studies on the intervention of physical exercise in FMS and PF of preschool children. However, our research aims to answer the question "which type of physical exercise can improve FMS and PF better in preschool children". We calculated the sample size to ensure sufficient power and planned to include samples exceeding the calculated value. And a 6-month follow-up is planned to determine the long-term effect of physical exercise on FMS and PF in preschool children.
- 5 kindergartens will be randomly allocated to 5 intervention groups. So, the study will compare the effects of 5 different kindergartens. I am not sure whether this could be problematic, as kids in different kindergartens could be very different, the quality of PA teachers in different kindergartens could also be different, potentially resulting in some differences in the tested results (FMS and PF), or even in the baseline results. It is more idea to compare in the same kindergarten, if possible, or to include more kindergartens (not only 5 kindergartens).
Response: Thank you very much for kind reminder. We have also considered this issue and developed a series of measures to reduce the differences in the quality of physical exercise caused by different interventionists. As mentioned in the manuscript, we recruit kindergartens in the same block and cooperate with the local education bureau. Therefore, these kindergartens have the same curriculum standards. In addition, the software and hardware conditions such as sports activity plan, outdoor sports equipment and teacher work plan are also similar to meet the inclusion conditions. After recruitment, we will train the interventionists (kindergarten teachers) before the formal intervention to ensure that they master the content of physical exercise that need to be implemented. And we will assign a researcher who familiar with the intervention content to each intervention group to supervise and assist interventionists to ensure the quality of the intervention. In addition, researchers will keep in touch with the interventionists to solve the relevant problems they encounter in time. You can check the "Delivery and fidelity check of interventions" section for details.
Minor notes
…and largely determines PA level… (line 49), it is weird that physical fitness determines physical activity? It is better to say that physical fitness is influenced by physical activity.
Response: Thanks for your suggestion, the related sentences have been adjusted to “support individual’s physical behavior, live and work with vigor”, please check line 49-50.
the effectiveness of interventions to improve FMS due to the quality of evidence…(line 66-67), this can be more specific and clear, so the readers know what do you mean “the quality of evidence”, e.g., what evidence showing poor quality of the results or…
Response: Thanks for your suggestion, the related sentences have been adjusted to “due to the high heterogeneity of comprehensive evidence and the lack of long-term follow up”, please check line 66-67.
(2) establish the comparative effectiveness of these physical…(line 82) seems to indicate the same thing as (1) determine the effectiveness of each of these physical exercise interventions...
Response: Thanks for your suggestion, the related sentences have been adjusted to “establish the differences between the effectiveness of different physical exercise pro-grams on preschool children’s FMS and PF”, please check line 95-96.

Reviewer 3 Report
The presented conception of the research is of significant importance for the development of the population of the youngest children. Monitoring and developing fitness and physical activity is very important in preschool age. The authors present the detailed assumptions of the research protocol. The clear and understandable way of presenting the assumptions of the protocol, eg figures 1 and 2, deserves recognition. The use of 30-minute lesson with the following parts: 5 minutes of warm-up, 20 minutes of core and 5 minutes of cooling down, seems right and adjusted to the age of pre-school children.
The all sections in paper are well written. The Authors have all ethics permissions.
The physical fitness tests seem to be well chosen, although I have some doubts about the use of the 20m shuttle-run test among 4 year olds. In my opinion, the use of a proressive test with a sound signal will be difficult for such young children, but I dont suggest that it is impossible.
I suggest, without knowing the distribution characteristics of the obtained data, not to write that the mean and standard deviation will be calculated. If the data is inconsistent with the normal distribution, the median and quartiles should be analyzed. The same goes for the statistical tests.
In my opinion, the article is written very well! The research concept does not raise my objections. Congratulations to the Authors.
Author Response
- The physical fitness tests seem to be well chosen, although I have some doubts about the use of the 20m shuttle-run test among 4 years old. In my opinion, the use of a proressive test with a sound signal will be difficult for such young children, but I dont suggest that it is impossible.
Response: Thank you very much for your concern. According to the relevant review (Ortega F B, Cadenas-Sánchez C, Sánchez-Delgado G, et al. Systematic Review and Proposal of a Field-Based Physical Fitness-Test Battery in Preschool Children: The PREFIT Battery. Sports Medicine, 2015, 45(4):533-555), the 20m shuttle-run test is widely used to test the cardiorespiratory fitness of children aged 4-6 years. And the 20 m shuttle run test has shown to be sensitive enough to detect changes induced by exercise-based interventions in 4-year-old children.
- I suggest, without knowing the distribution characteristics of the obtained data, not to write that the mean and standard deviation will be calculated. If the data is inconsistent with the normal distribution, the median and quartiles should be analyzed. The same goes for the statistical tests.
Response: Thank you very much for your suggestion. We have adjusted and added more considerations if the data does not conform to the normal distribution: “non-parametric test will be used when the data does not conform to the normal distribution”, please check line 351, 356-357.

Round 2
Reviewer 2 Report
The authors have addressed my concerns.